# Colon Bioaccessibility under In Vitro Gastrointestinal Digestion of Different Coffee Brews Chemically Profiled through UHPLC-Q-Orbitrap HRMS

**DOI:** 10.3390/foods10010179

**Published:** 2021-01-17

**Authors:** Luigi Castaldo, Luana Izzo, Alfonso Narváez, Yelko Rodríguez-Carrasco, Michela Grosso, Alberto Ritieni

**Affiliations:** 1Department of Pharmacy, University of Naples “Federico II”, 49 Domenico Montesano Street, 80131 Naples, Italy; luana.izzo@unina.it (L.I.); alfonso.narvaezsimon@unina.it (A.N.); alberto.ritieni@unina.it (A.R.); 2Laboratory of Food Chemistry and Toxicology, Faculty of Pharmacy, University of Valencia, Av. Vicent Andrés Estellés s/n, Burjassot, 46100 València, Spain; yelko.rodriguez@uv.es; 3Department of Molecular Medicine and Medical Biotechnology, School of Medicine, University of Naples “Federico II”, CEINGE-Biotecnologie Avanzate, 80131 Naples, Italy; michela.grosso@unina.it; 4Staff of UNESCO Chair on Health Education and Sustainable Development, “Federico II” University, 80131 Naples, Italy

**Keywords:** polyphenols, coffee, chlorogenic acids, in vitro gastrointestinal digestion, bioaccessibility

## Abstract

Coffee represents one of the most traditionally consumed beverages worldwide, containing a broad range of human health–related compounds. According to previous studies, regular coffee consumption may display protective effects against colorectal cancer and other chronic diseases. The main goal of this research was to evaluate the bioaccessibility of phenolic content and variation in antioxidant capacity of three different types of coffee brews after simulated gastrointestinal digestion. This would allow to elucidate how antioxidant compounds present in coffee may exert their effect on the human body, especially in the colonic stage. Moreover, the content of bioactive compounds namely chlorogenic acids (CGAs, *n* = 11) and caffeine was also assessed throughout ultra-high-performance liquid chromatography followed by high-resolution Orbitrap mass spectrometry (UHPLC-Q-Orbitrap HRMS). The three main isomers of caffeoylquinic acid constituted the highest fraction of CGAs present in the samples, accounting for 66.0% to 70.9% of total CGAs. The bioaccessibility of coffee polyphenols significantly increased in digested samples from 45.9% to 62.9% at the end of the colonic passage, compared to the non-digested samples. These results point to the colonic stage as the major biological site of action of the active antioxidant coffee compounds.

## 1. Introduction

Coffee represents one of the most traditionally consumed beverages worldwide, so its preparation method and consumption vary according to consumer’s preferences, traditions, and social behavior, among other factors [1]. Over the last decade, a large number of studies have supported the apparent beneficial properties of habitual coffee consumption for human health [2]. Several studies concluded that coffee contains high concentrations of bioactive compounds that may display a positive influence on human health, including polyphenols, alkaloids, fibers, and dark-colored compounds formed during the roasting process, namely melanoidins [3,4].

Polyphenols are ubiquitous compounds formed as a result of the secondary metabolism of plants that cover a broad range of necessities in their home plant and whose biological activity has also been observed in humans [5]. Scientific evidence links a frequent intake of polyphenols to health benefits, preventing several age-related diseases, such as cancer or cardiovascular disease [6]. In coffee, the most abundant polyphenolic compounds are represented by chlorogenic acids (CGAs). These important compounds are known to have a strong antioxidant property, able to promote benefits to human health [7]. Structurally, CGAs consist of a skeleton of quinic acid esterified with a hydroxycinnamic acid such as *p*-coumaric, caffeic, cinnamic, or ferulic acids, with caffeoylquinic (CQA) and feruloylquinic (FQA) being the most relevant CGA fractions in coffee. 

An ever-expanding amount of literature links regular coffee intake to beneficial effects against some chronic diseases such as type 2 diabetes, obesity, Parkinson’s disease, and several typologies of cancer [8]. In this line, recent studies have showed an apparent relation between moderate coffee consumption and a significant lower probability of colorectal cancer, even in a dose-dependent way [9,10]. Furthermore, reduction of the risk of colorectal cancer was also reported for decaffeinated coffee, pointing to other components besides caffeine as being responsible for increasing colorectal protection [11]. Several plausible biological processes have been postulated in order to explain the benefits of coffee on reducing colorectal cancer risk [12], and polyphenols with important antioxidant capacity may play a key role in explaining this phenomenon [13]. Moreover, melanoidins represent another source of CGAs in coffee. These compounds appear in the late step of the Maillard reaction between carbohydrates and proteins during the roasting, resulting in a polymeric structure that can also include polyphenolic acids, such as CGAs. Melanoidins escape digestion to be metabolized by different microbial species during their passage through the colon, resulting in the release of antioxidant, low-molecular-weight substances attached to them [14]. Several studies have indicated that coffee melanoidins could be responsible for the antioxidant capacity shown by coffee brews, due to the CGAs incorporated [15].

According to Waisundara et al. [16], bioaccessibility depends on the amount of a food constituent that is present in the gastrointestinal tract, becoming available for absorption as a consequence of its release from the solid matrix. The enzymatic activity, temperature, pH, and bacterial microflora represent the most relevant factors involved in the release of native polyphenols embedded in the food matrix [17]. In vitro digestion protocols represent an useful alternative to in vivo assays in order to evaluate the bioaccessibility and structural changes of macronutrients in foodstuffs [18]. Nevertheless, the bioaccessibility and antioxidant capacity of the polyphenolic fraction contained in different coffee brews have been barely investigated after simulated gastrointestinal digestion, and limited data are currently available about the metabolic fate of coffee bioactive compounds in the large intestine [19,20,21]. However, the evaluation of the polyphenols’ bioaccessibility during the digestion process seems to be crucial to understand the potential beneficial effects on human health.

The brewing methods exert a significant influence on the polyphenol extraction, caffeine content, and antioxidant activity of the final coffee cup [22]. Although espresso is the most usual coffee brewing method in Europe, instant coffee is becoming a popular alternative, whereas Americano coffee represents one of the most common coffees globally sold. 

Hence, this study focused on providing useful information about the colon bioaccessibility of active compounds and variation in antioxidant capacity of three types of coffee brews following simulated gastrointestinal digestion. This would allow elucidating how antioxidant compounds present in coffee may exert their effect in the human body, especially, in the colon stage. Moreover, the bioactive compounds such as CGAs (*n* = 11) and caffeine contained in 30 samples belonging to three different types of coffee brews namely espresso, instant, and Americano coffee brews were also evaluated throughout ultra-high-performance liquid chromatography followed by high-resolution Orbitrap mass spectrometry (UHPLC-Q-Orbitrap HRMS).

## 2. Materials and Methods 

### 2.1. Reagents and Materials

Ethanol (EtOH), methanol (MeOH), water (LC-MS grade), hydrochloric acid (HCl), and formic acid (FA) (analytical grade) were obtained from Carlo Erba Reagents (Milan, Italy), whereas deionized water (<18 MΩ cm resistivity) was obtained from a Milli-Q water purification system (Millipore, Bedford, MA, USA). Potassium chloride (KCl), 2′2-azino-bis-3-ethylbenzthiazoline-6-sulfonic acid (ABTS), 2,4,6-tris(2-pyridyl)-1,3,5 triazine (TPTZ), 6-hydroxy-2,5,7,8-tetramethylchromane-2-carboxylic acid (commonly called Trolox), 1,1-diphenyl-2-picrylhydrazyl (DPPH), Folin–Ciocalteu reagent, hydrochloric acid, sodium acetate, sodium hydroxide (NaOH), monosodium phosphate (NaH_2_PO_4_), sodium bicarbonate (NaHCO_3_), potassium thiocyanate (KCNS), calcium chloride dihydrate (CaCl_2_·2H_2_O), ferric chloride (FeCl_3_), sodium chloride (NaCl), sodium sulfate (Na_2_SO_4_), sodium carbonate (Na_2_CO_3_), and potassium persulfate (K_2_S_2_O_8_) were purchased from Sigma-Aldrich (Milan, Italy). Standards (purity >98%) were obtained as follows: gallic acid, 3,4-dicaffeoylquinic acid, 4-caffeoylquinic acid, and caffeine were acquired from Sigma-Aldrich (Milan, Italy). To simulate gastrointestinal digestion, the following enzymes were used: pepsin (≥2500 U/mg solid) from porcine gastric mucosa pancreatin (8 × United States pharmacopeia, USP) from porcine pancreas, α-amylase (1000–3000 U/mg solid) from human saliva, Viscozyme L and bacterial protease from *Streptomyces griseus* (known as Pronase E, ≤3.5 U/mg solid), all obtained from Sigma-Aldrich (Milan, Italy).

### 2.2. Sampling

All samples were acquired from Italian local markets. Espresso (*n* = 10) and Americano samples (*n* = 10) were delivered as vacuum-packed, medium-roasted coffee beans, whereas spray-dried, instant coffee samples (*n* = 10) were obtained in powder/granule form. Colombian Arabica, medium-roasted beans for espresso coffee were finely ground with a coffee grinder (DeLonghi, KG79, Treviso, Italy), and the obtained powder was passed through a 0.4 mm sieve.

### 2.3. Coffee Preparation

This study analyzed three different brewing methods chosen based on their wide consumption all over the world, namely espresso, Americano, and Instant coffee. 

The espresso brews were prepared as described by Martinez et al. [23]. Briefly, 7 g of coffee powder and hot distilled water using an espresso machine (DeLonghi, Icona Vintage, Treviso, Italy) working with a pressure of 15 bar and water temperature of 90 °C as declared by the manufacturer. Then, 25 mL of coffee brew was collected into a volumetric tube. Americano coffee brew was prepared as described by Liu et al. [24]. Briefly, 200 mL of distilled water was added to the single-shot espresso and thoroughly mixed. Instant coffee brews were prepared by pouring 200 mL of boiling distilled water over 6 g of instant coffee powder and stirring until dissolved as indicated by the manufacturer. Each coffee preparation was obtained at least 5 times and stored at −18 °C until analysis.

### 2.4. Content of High-Molecular-Weight Melanoidins 

High-molecular-weight melanoidins (HMWM) were quantified according to the previously described procedure [25]. Briefly, 4 mL of coffee brew properly diluted with demineralized water was subjected to ultrafiltration using an Amicon Ultra-4 (Millipore, Italy) at 7500× *g* for 70 min at 4 °C, equipped with a 10 kDa nominal molecular mass cutoff membrane. The retentate was filled with 4 mL of water and washed five times for reaching a proper separation of the low-molecular-weight compounds. Quantification of total HMWM was performed considering the weight of the freeze-dried retentate obtained after dialysis, and results were expressed as mg/mL of sample. 

### 2.5. Ultra-High-Performance Liquid Chromatography and Orbitrap High-Resolution Mass Spectrometry Analysis

Chromatographic separation was carried out using an UHPLC (Dionex UltiMate 3000, Thermo Fisher Scientific, Waltham, MA, USA) prepared with a quaternary pump with a pressure tolerance of 1250 bar, a degassing system, and an autosampler device. Separation of CGAs and caffeine occurred within a thermostated (T = 25 °C) Kinetex 1.7 µm F5 (50 × 2.1 mm, Phenomenex, Torrance, CA, USA) column.

The eluents consisted of water (phase A) and methanol (phase B), both containing 0.1% of formic acid. The separation gradient started with 0% B, and then increased to 40% B in 1 min, to 80% during, and then rose up again reaching 100% B in 3 min and maintained over 4 min. Finally, the gradient returned to 0% B in 2 min and was maintained over 2 min for column re-equilibration. The injection volume was 1 µL with a flow rate of 0.5 mL/min, whereas the total gradient time was 13 min.

Mass spectrometry was performed using a Q-Exactive mass spectrometer (Thermo Fischer Scientific, Waltham, MA, USA) prepared with an electrospray (ESI) source that allowed simultaneous operation in negative and positive ionization modes due to the fast polarity switching mode. The acquisitions were based on two scan events: all ion fragmentation (AIF) and full-ion MS. The ion source parameters were set as follows: capillary temperature 320 °C, spray voltage 3.5 kV, S-lens RF level 60, auxiliary gas 3, sheath gas pressure 18, and auxiliary gas heater temperature 350 °C. The full MS mode experiment was carried out considering the following parameters: scan range 80–1200 *m*/z, resolution power of 70,000 full width at half-maximum (FWHM) (defined for *m/z* 200), automatic gain control (AGC) target 1 × 10^6^, scan rate set at 2 scan/s, and injection time set to 200 ms. AIF scan conditions were as follows: maximum injection time of 200 ms, resolution set at 17,500 FWHM, scan time of 0.10 s, scan range 80–120 *m*/*z*, AGC target 1 × 10^6^, retention time of 30 s, and an isolation window of 5 *m*/*z*. The collision energy was optimized considering values in the range 10–60 eV, when the parent ion represented 10% of the relative intensity whereas the product ion was held at 90% of relative intensity.

Identification was carried out considering exact mass measurements with a mass error <5 ppm. Data treatment was performed through Quan/Qual Browser Xcalibur software, v 3.1.66.19 (Xcalibur, Thermo Fischer Scientific, Waltham, MA, USA). 

### 2.6. In Vitro Simulated Gastrointestinal Digestion

Coffee brews samples were brought under successive oral, gastric, and intestinal in vitro digestion, following an harmonized procedure recently created by the COST action INFOGEST network [26]. Simulated digestion fluids, namely gastric fluid (SGF), salivary fluid (SSF), and intestinal fluid (SIF) were built considering a previously described procedure [27] and showed in Appendix A, Appendix A.

Briefly, 3.5 mL of SSF (T = 37 °C) was added to 5 mL of sample and mixed. Next, 25 µL of 0.3 M calcium chloride, 0.5 mL of α-amylase solution (75 U/mL), and 975 µL of water were added and mixed. A solution 1 M HCl was added to reduce the pH of the solution to 7, and the mixture was incubated at 37 °C for 2 min in an orbital shaker bath at 100× *g*.

Then, for simulating gastric conditions, 1.6 mL pepsin solution (2000 U/mL), 7.5 mL SGF, 695 µL of water, and 5 µL 0.3 M calcium chloride were added and thoroughly mixed. Next, HCl 1 M was used to decrease the pH of the solution to 3, and the mixture was incubated for 120 min at 37 °C in an orbital shaker bath at 50× *g*.

Afterward, to recreate the intestinal stage, 11 mL SIF, 5 mL pancreatin solution (100 U/mL of trypsin activity), 2.5 mL bile salt solution (65 mg/mL), 1.3 mL of water, and 40 µL of 0.3 M calcium chloride were added. After that, the solution was thoroughly mixed, and 1 M NaOH was added to increase the pH of the mixture to 7. The solution was incubated at 37 °C for 120 min in an orbital shaker bath at 100× *g*, and then, centrifuged for 10 min at 37 °C at 4900× *g*.

Finally, to determine the bioaccessibility of the polyphenolic fraction and the antioxidant capacity in the colon stage, the remaining pellet was subjected to the previously described procedure [27]. First, 5 mL of 1 mg/mL Pronase E solution was added, and the pH was readjusted to 8 using NaOH 1M, simulating the action of the gut microbiota. The mixture was then incubated at 37 °C for 60 min in an orbital shaker bath at 100× *g*. Next, the mixture was treated with 150 µL of Viscozyme L. The pH was readjusted to 4 and incubated at 37 °C for 16 h and centrifuged at 4900× *g* for 10 min at 24 °C.

At each digestion stage, aliquots (*n* = 3) of the supernatant were collected, freeze-dried, and then stored at −18 °C until further analysis.

### 2.7. Determination of the Antioxidant Activity

The antioxidant activity of the different coffee brews subjected to the in vitro gastrointestinal digestion and non-digested samples was determined and spectrophotometrically compared through three different assays. Results were expressed as millimoles of Trolox equivalents (TEs) per mL of sample. Measurements were performed in triplicate.

#### 2.7.1. ABTS Assay

The ABTS scavenging capacity assay was carried out according to the procedure described in a previously published article [28], with slight adaptations. Briefly, 44 µL of aqueous potassium persulfate (2.45 mM) was added to aqueous ABTS (7 mM), and thoroughly mixed. The mixture was kept in dark conditions at room temperature for 16 h. Afterward, the scavenging capacity was evaluated by diluting the ABTS solution in ethanol by adjusting the absorbance of the ABTS radical solution to 0.70 ± 0.02 at 734 nm. Next, 100 µL of properly diluted sample was added to 1 mL of ABTS radical working solution, and after 3 min, the absorbance was measured. 

#### 2.7.2. DPPH Assay

The DPPH free-radical scavenging capacity of the samples was evaluated through the methodology described by Dini et al. [29] slightly modified. In short, methanolic DPPH (1 mg in 2.5 mL) was diluted using methanol until reaching an absorbance value of 0.90 ± 0.02 at 517 nm. Afterward, 200 µL of properly dilute sample was mixed with 1 mL of DPPH radical working solution and, after 10 min, the decrease in the absorbance at 517 nm was monitored.

#### 2.7.3. Ferric Reducing/Antioxidant Power (FRAP) Assay

Measurements from the FRAP assay were obtained following the methodology reported by Rajurkar et al. [30] with slight modifications. In brief, the FRAP solution was obtained by mixing 1.25 mL of TPTZ solution (10 mM) in HCl (40 mM), 1.25 mL of ferric chloride solution (20 mM) in H_2_O, and 12.5 mL of acetate buffer (0.3 M, pH 3.6). Next, 150 µL of the properly diluted sample were added to 2.85 mL of FRAP solution. After 4 min, the absorbance was immediately measured at 593 nm.

### 2.8. Determination of Total Phenolic Content

Total phenolic content (TPC) was evaluated in accordance with the methodology previously described by Izzo et al. [31] with some adaptations. Briefly, 125 µL of the properly diluted sample were mixed with 125 µL of the Folin–Ciocalteu reagent 2 N and 500 µL of deionized water for later incubation at room temperature for 6 min. Afterward, 1 mL of deionized water and 1.25 mL of a 7.5% sodium carbonate solution were added. The absorbance after 90 min at 750 nm was monitored. Results were reported as mg of gallic acid equivalents (GAEs) per mL of sample.

### 2.9. Statistics and Data Analysis

All the measurements were carried out in triplicate and the results expressed as the average value ± standard deviation (SD). Statistical analysis was carried out using STATA 12 software (STATA Corp LP, College Station, TX, USA). Tukey’s test was used for evaluating statistical differences among groups with a significance level of *p* < 0.05. The correlation coefficients were calculated using Pearson’s method.

## 3. Results

### 3.1. HMWM Content

The amounts of HMWM obtained in the different coffee brews samples are summarized in Appendix A, Appendix A. The highest content in HMWM resulted from espresso brews, reaching 6.23 mg/mL, whereas instant coffee showed a similar quantity (5.78 mg/mL). On the contrary, the quantified HMWM in the Americano brew (0.71 mg/mL) were between 8 and 9 times lower compared to that in the other brews.

### 3.2. Identification of Caffeine and CGAs in the Analyzed Coffee Brew Samples through UHPLC-Q-Orbitrap HRMS

A total of 11 different CGAs were identified in the analyzed coffee brew samples through the combination of MS and MS/MS spectra. Data for ion assignment, retention time (RT), measured mass (*m*/*z*), theoretical mass (*m*/*z*), and accuracy are presented in Table 1. 

All the studied CGAs exhibited better fragmentation patterns in negative ion mode, whereas caffeine was detected in positive ion mode. The untargeted compounds’ identification was enabled by the full-scan screening through a retrospective data analysis. The untargeted analytes’ structural characterization relied on the MS/MS spectrum interpretation (Appendix A in Appendix A), elemental composition assignment, and accurate mass measurement. The structural isomers CQA (*m*/*z* 353.08780), diCQA (*m*/*z* 515.11950), *p*-coumaroylquinic (*p*CoQA; *m*/*z* 337.09289), caffeoyl-feruloylquinic and feruloyl-caffeoylquinic acids (CFQA and FCQA; *m*/*z* 529.13245) were identified by comparing their fragmentation pattern with information previously reported in the literature [25] and by comparing the retention times of standards with those of the peaks. The chromatographic separation of the analyzed compounds was performed in a total run time of 13 min. The 4-FQA and 5-FQA were semi-quantified together due to insufficient chromatographic separation.

### 3.3. Quantification of Caffeine and CGAs in the Analyzed Coffee Brew Samples

The predominant CGAs and caffeine were determined through UHPLC-Q-Orbitrap HRMS analysis. Calibration curves built with eight concentration levels were used in the quantitative determination of target analytes, and the correlation coefficients obtained were >0.990. A representative standard from the same group was used for the semi-quantification purpose, the compounds with no standard available were 3 and 5-*p*CoQA; 3, 4 and 5-FQA; 3 and 5-CQA; and 4,5-CFQA and 3,4-FCQA isomers.

As shown in Table 2, twelve analytes were identified, quantified, or semi-quantified in the different coffee brews samples. Among the CGAs, CQAs were the most abundant investigated compounds found in the samples, ranging from 66.04% (instant coffee) to 70.92% (espresso) of total CGAs with a concentration level ranging between 17.69 (Americano) and 158.31 mg/100 mL (espresso). Moreover, 5-CQA was the most predominant CQA in the assayed coffee brew samples, ranging from 12.34 (Americano) to 110.14 mg/100 mL (espresso), with an average content of 53.23 mg/100 mL for all coffee brews.

In the coffee brew samples investigated here, FQAs represented from 20.20% (Americano) to 26.48% (instant coffee) of total CGAs with a concentration level ranging between 18.92 and 24.81 mg/100 mL.

As shown in Table 2, among *p*CoQAs, 5-*p*CoQA was semi-quantified at concentrations significantly higher than the other investigated *p*CoQAs, ranging from 0.84 (Americano) up to 5.78 mg/100 mL (espresso), with an average content of 2.57 mg/100 mL.

Regarding the diCQAs group, 3,4-diCQA was the most predominant compound, at levels ranging from 0.72 to 6.47 mg/100 mL, with an average content of 2.98 mg/100 mL.

Moreover, isomers 3,4-FCQA and 4,5-CFQA proved to be the less relevant CGAs, being measured at average values of 0.12 and 0.34 mg/100 mL for all assayed samples, respectively.

Apart from CGAs, caffeine was quantified in the here-analyzed coffee brews samples. Espresso coffee brew samples showed higher caffeine content (331.59 mg/100 mL) when compared to Americano (41.82 mg/100 mL) and instant coffee (124.05 mg/100 mL) brews, as shown in Table 2.

### 3.4. In Vitro Bioaccessibility of Coffee Polyphenols

Simulated gastrointestinal digestion was used to evaluate the coffee polyphenol bioaccessibility. TPC content was measured using the Folin–Ciocalteu assay in the various stages of the gastrointestinal digestion. To obtain an overview of their bioaccessibility, a comparison with the non-digested samples was done. Table 3 shows the mean values of TPC and the percentage of increase (not-digested vs. digested stage) in each phase of the simulated gastrointestinal digestion. Among the non-digested samples, the highest TPC value resulted from the espresso coffee brew sample with a mean value of 19.8 mg GAE/mL and, the lower value was shown by the Americano coffee brew sample (2.1 mg GAE/mL). On the other hand, the results showed a significant improvement (*p*-value <0.05) in the polyphenol bioaccessibility after all stages of gastrointestinal digestion. Moreover, the percentage of increase in TPC after simulated gastrointestinal digestion ranged from 10.0% up to 49.7%, and the highest values were displayed after the colonic fermentation (considered as Pronase stage plus Viscozyme L stage).

### 3.5. Antioxidant Capacity of the Coffee Brew Samples during Simulated Gastrointestinal Digestion

The antioxidant capacity of the different coffee brews samples subjected to the gastrointestinal digestion and non-digested samples was determined and compared through three different methods (ABTS, DPPH, and FRAP); data were expressed as millimoles of TE per 100 mL of coffee brew. Table 4 shows the results (the mean value and SD) for all analyzed samples at each in vitro digestion stage.

As shown in Table 4, among the non-digested samples, the espresso coffee brew samples showed the highest antioxidant activity, followed by instant and Americano coffee brews in all the performed procedures. Concerning the antioxidant activity measured after the simulated gastrointestinal digestion, digested coffee brew samples showed an increase in all stages than the initial ones (*p*-value <0.05). Moreover, the antioxidant activity during gastrointestinal digestion experimented a fourfold increase, an eightfold increase, and a two-fold increase in ABTS, DPPH, and FRAP assays, respectively. In particular, the colonic stage showed the highest increase in all assayed samples (considered as PS plus VS). Comparing the antioxidant activity values obtained after ABTS, DPPH, and FRAP assays after the simulated gastrointestinal digestion against their corresponding TPC evaluated by Folin–Ciocalteu, positive correlations were observed, as shown in Table 5.

## 4. Discussion

The main goal of this research was to evaluate the bioaccessibility of the polyphenolic fraction of coffee and the antioxidant activity after in vitro gastrointestinal digestion and to establish the polyphenolic profile through UHPLC Q-Orbitrap mass spectrometry in order to characterize the chemical composition of the most common coffee brews.

Overall, the data indicate that the analyzed coffee brews may represent a great source of important active molecules, such as a high content of CGAs, caffeine, and melanoidins. The brewing procedure affects the concentration of active molecules present in a cup of coffee. Concerning the CGAs in assayed coffee brew samples, the concentrations found in coffee espresso were higher when compared to that in other coffee brew methods assayed. These results may be explained by the use of the higher proportion of ground coffee per water volume; 0.28 g/mL for espresso and 0.03 g/mL for both Americano and instant coffee, respectively. In general, the contents of CGAs in different coffee brews present largely variability in data reported in literature, from 26 to 1141 mg/100 mL, being highly dependent on the variety of coffee, the roast degrees, and the brewing method used [32]. Nevertheless, the most studied CGAs in coffee brews are the three main CQA isomers [33], whereas FQAs and diCQAs have been barely investigated in these types of coffee brews. In this study, 11 different CGAs were simultaneously monitored, with FQAs and CQAs being the most commonly found in all the analyzed coffee brews. Several scientific studies have reported that the CGAs are responsible for important biological effects in maintaining health status and preventing human diseases, which might be due to the ability of CGAs to modulate lipid and glucose metabolism, helping to prevent several disorders including colorectal cancer, obesity, cardiovascular disease, and diabetes as well [34,35,36,37]. Moreover, our findings highlighted that the espresso coffee brew sample showed higher caffeine content per mL of beverage (3.3 mg/mL) when compared to other analyzed coffee brews. These results reflected a two-to-three times increase in caffeine compared to instant coffee. The concentrations found in the analyzed samples are in line with the levels reported in the literature [22,38]. The higher concentration in the mentioned alkaloid found in espresso coffee is balanced by the limited volume of serving cup (15 to 50 mL) [39]. Nevertheless, taking into account the traditional serving size of a coffee cup, instant coffee showed the highest content of CGAs and caffeine compared to espresso and Americano coffee.

On the other hand, the evaluation of colonic bioaccessibility and the antioxidant capacity of the coffee polyphenols after simulated gastrointestinal digestion was the main aim of this study. Since the different classes of antioxidants can react in different mechanisms, three distinct methods were selected in this work in order to obtain a complete antioxidant profile of the active molecules present in the analyzed coffee brews [40]. The results showed that coffee polyphenol bioaccessibility and the antioxidant capacity significantly increased after the colonic stage in all analyzed coffee brews. Recently, similar results have also been reported in literature, where the aqueous coffee silverskin extracts, rich in melanoidins and CGAs, showed the highest percentage of increase in terms of bioaccessibility and antioxidant capacity after colonic stage [25]. Similar outcomes were obtained by Annunziata et al. [41], who demonstrated that both colon bioaccessibility and the antioxidant activity of polyphenols from tea were significantly higher compared to the other digestion phases. However, Jilani et al. [42] showed that the black and green tea polyphenols’ bioaccessibility decreased in the intestinal phase, while the gut microbiota enhanced the total antioxidant activity only in black tea samples. 

Campos-Vega et al. [43] also demonstrated that colonic fermentation elevates the antioxidant capacity of polyphenols from spent coffee when compared with their respective non-digested samples due to the high release of phenolic compounds. The antioxidant activity was assessed with 2 of 3 assays used in the present work namely the DPPH and ABTS test. These findings suggest that coffee’s active molecules could be metabolized by gut microbiota, releasing analytes with higher antioxidant potential, which may potentially increase their beneficial effects. Moreover, the quantification of the melanoidins, one of the most relevant compounds found in coffee brews, was necessary to fully explain the bioaccessibility of phenolic compounds and the variation in the antioxidant capacity of coffee brews after simulated gastrointestinal digestion. In fact, the increased bioaccessibility and antioxidant activity observed in assayed samples appeared to be attributable to the coffee melanoidins. According to the recent publications, coffee melanoidins escape digestion, similar to dietary fibers, and during the colonic stage they can become substrates for the gut microbiota, releasing antioxidant active molecules linked to them [44,45,46]. The released phenolics, especially CGAs, incorporated during the roasting of coffee beans, may exert a local effect in protection against colorectal cancer and provide important health benefits after absorption via epithelium cells [43]. Moreover, the results showed a strong positive correlation among TPC values and data obtained from ABTS, DPPH, and FRAP assays evaluated after the simulated gastrointestinal digestion, highlighting that the assayed procedures provide reliable information on the antioxidant capacity of molecules released after the simulated gastrointestinal digestion. The limitations of our study are mainly related to the diversity of coffee beans used in coffee preparation. Although the espresso and Americano coffee brews were prepared with the same coffee bean samples, the instant coffee samples used in the present study were acquired directly in powder/granule form. This choice was due to the inability to prepare reliable samples from coffee beans that could be comparable to commercially available products in light of the no-replicate industrial procedures.

## 5. Conclusions

This study provided useful information on the predominant active molecules including CGAs (*n =* 11) and caffeine in three different types of coffee brews through UHPLC-Q-Orbitrap spectrometry measurement. Notably, caffeoylquinic and feruloylquinic acids were the most abundant CGAs in the coffee brews. Moreover, the results highlighted that, after the simulated gastrointestinal digestion, the bioaccessibility and antioxidant activity of coffee polyphenols increased when compared with the non-digested samples, and the colonic stage may represent the major biological site of action of antioxidant coffee compounds. Melanoidins contained in coffee could be metabolized by the microbiota in the colon, generating several metabolites with higher antioxidant activity and more beneficial effects. Polyphenolic compounds released from melanoidins due to the action of colonic microflora could explain these outcomes. Nevertheless, more in-depth knowledge is needed to elucidate the several biotransformations namely methylation, deglycosylation, and other catabolic breakdown that involve the major CGAs during the gastrointestinal process and evaluate their actions on human health.

## Figures and Tables

**Table 1 foods-10-00179-t001:** UHPLC-MS parameters of the investigated analytes (*n* = 12).

Compound	Adduct	Chemical	RT	Measured	Theoretical	Accuracy
	Ion	Formula	(min)	Mass *(m/z)*	Mass *(m*/*z)*	(Δ mg/kg)
5-CQA	[M−H]^−^	C_16_H_18_O_9_	3.11	353.08790	353.08780	0.03
3-*p*CoQA	[M−H]^−^	C_16_H_18_O_8_	3.13	337.09232	337.09289	−1.69
3-FQA	[M−H]^−^	C_17_H_20_O_9_	3.18	367.10303	367.10346	−1.17
**Caffeine ***	[M+H]+	C_8_H_10_N_4_O_2_	3.25	195.08757	195.08765	−0.41
**4-CQA ***	[M−H]^−^	C_16_H_18_O_9_	3.31	353.08768	353.08780	−0.34
5-*p*CoQA	[M−H]^−^	C_16_H_18_O_8_	3.32	337.09290	337.09289	0.03
**3,4-diCQA ***	[M−H]^−^	C_25_H_24_O_12_	3.36	515.12103	515.11950	2.97
4+5-FQA	[M−H]^−^	C_17_H_20_O_9_	3.38	367.10309	367.10346	−1.01
4,5-CFQA	[M−H]^−^	C_26_H_26_O_12_	3.43	529.13495	529.13245	4.72
3-CQA	[M−H]^−^	C_16_H_18_O_9_	3.51	353.08762	353.08780	−0.51
3,5-diCQA	[M−H]^−^	C_25_H_24_O_12_	3.56	515.11993	515.11950	0.83
3,4-FCQA	[M−H]^−^	C_26_H_26_O_12_	3.67	529.13247	529.13245	0.04

* Compounds in bold were quantified using the real standard.

**Table 2 foods-10-00179-t002:** Chlorogenic acids (CGAs) and caffeine content in the analyzed coffee brew samples. Results expressed as the average value ± standard deviation. Differences between groups were statistically analyzed with Tukey’s test; *p* < 0.05 was considered significant.

Compound	Espresso	Americano	Instant Coffee
	Average (mg/100 mL) ± SD	Average (mg/100 mL) ± SD	Average (mg/100 mL) ± SD
3-CQA	19.09 ± 0.49	2.12 ± 0.01	8.47 ± 0.34
4-CQA	29.09 ± 0.08	3.23 ± 0.03	11.17 ± 0.42
5-CQA	110.14 ± 2.12	12.34 ± 0.09	37.21 ± 0.61
3-*p*CoQA	0.08 ± 0.01	0.01 ± 0.00	0.01 ± 0.00
5-*p*CoQA	5.78 ± 0.21	0.84 ± 0.01	1.08 ± 0.01
3-FQA	2.74 ± 0.22	0.32 ± 0.01	1.43 ± 0.01
4+5-FQA	42.83 ± 0.51	4.76 ± 0.05	21.36 ± 0.71
3,4-diCQA	6.47 ± 0.21	0.72 ± 0.02	1.75 ± 0.01
3,5-diCQA	3.41 ± 0.11	0.38 ± 0.01	1.58 ± 0.01
3,4-FCQA	0.22 ± 0.01	0.02 ± 0.00	0.11 ± 0.00
4,5-CFQA	0.60 ± 0.01	0.07 ± 0.00	0.37 ± 0.00
Caffeine	331.59 ± 5.12	41.82 ± 0.63	124.05 ± 3.12
Total CGAs	220.45 ± 9.50	24.81 ± 0.03	84.56 ± 0.26

**Table 3 foods-10-00179-t003:** Total phenol content (TPC) of assayed samples (not-digested vs. digested stages).

Samples	TPC in Espresso		TPC in Americano		TPC in Instant Coffee	
	mg GAE/mL ± SD	%	mg GAE/mL ± SD	%	mg GAE/mL ± SD	%
Not digested	19.8		2.1		12.3	
Digestion Stage						
Oral satge	22.2 ± 0.2	12.2	2.4 ± 0.1	14.7	15.1 ± 0.3	23.0
Gastric stage	24.1 ± 0.3	22.0	2,6 ± 0.1	24.5	15.9 ± 0.3	29.1
Duodenal stage	27.0 ± 0.4	36.7	2.9 ± 0.2	41.8	16.2 ± 0.4	31.4
Pronase (Colonic phase I)	28.5 ± 0.4	43.9	3.1 ± 0.2	49.7	16.3 ± 0.4	32.7
Viscozyme L (Colonic phase II)	21.7 ± 0.3	10.0	2.3 ± 0.1	13.2	13.9 ± 0.3	13.2
Total colonic stage	50.2 ± 0.4	153.9	5.4 ± 0.2	162.9	30.33 ± 0.4	145.9

%: Percentage of increase in TPC (not-digested vs. digested stage). Differences between groups were statistically analyzed with Tukey’s test; *p* < 0.05 was considered significant.

**Table 4 foods-10-00179-t004:** Antioxidant capacity evaluated by ABTS, DPPH, and ferric reducing/antioxidant power (FRAP) procedures of not-digested and digested samples.

	ABTS mmol TE/100 mL ± SD			DPPH mmol TE/100 mL ± SD			FRAP mmol TE/100 mL ± SD		
	Espresso	Americano	Instant Coffee	Espresso	Americano	Instant Coffee	Espresso	Americano	Instant Coffee
Not digested	5.0 ± 0.4	0.6 ± 0.1	1.2 ± 0.1	4.0 ± 0.3	0.5 ± 0.1	0.9 ± 0.2	4.6 ± 0.3	0.6 ± 0.1	1.1 ± 0.2
Digestion Stage									
Oral satge	5.4 ± 0.3	0.8 ± 0.1	1.3 ± 0.1	5.6 ± 0.4	0.6 ± 0.1	1.4 ± 0.2	5.1 ± 0.2	1.9 ± 0.3	1.3 ± 0.1
Gastric stage	6.4 ± 0.5	1.9 ± 0.2	1.3 ± 0.1	5.5 ± 0.4	1.6 ± 0.2	1.3 ± 0.3	6.0 ± 0.4	0.7 ± 0.1	1.3 ± 0.1
Duodenal stage	6.6 ± 0.5	1.6 ± 0.3	1.4 ± 0.1	5.9 ± 0.3	0.7 ± 0.1	1.2 ± 0.2	6.3 ± 0.5	0.7 ± 0.1	1.4 ± 0.1
Pronase (Colonic phase I)	7.4 ± 0.4	1.0 ± 0.2	6.7 ± 0.4	6.3 ± 0.3	0.7 ± 0.1	8.3 ± 0.6	6.3 ± 0.4	0.7 ± 0.2	1.4 ± 0.2
Viscozyme L (Colonic phase II)	6.0 ± 0.4	0.8 ± 0.1	6.2 ± 0.4	4.3 ± 0.3	0.6 ± 0.1	5.2 ± 0.4	5.3 ± 0.4	0.8 ± 0.1	1.3 ± 0.1
Total colonic stage	13.5 ± 0.4	1.8 ± 0.2	12.9 ± 0.4	10.7 ± 0.3	1.3 ± 0.1	13.5 ± 0.5	11.6 ± 0.4	1.5 ± 0.2	2.7 ± 0.2

Differences between groups were statistically analyzed with Tukey’s test; *p* < 0.05 was considered significant.

**Table 5 foods-10-00179-t005:** Correlation between TPC and antioxidant activity data. The correlation coefficients were calculated Pearson’s method.

Assay	Oral Stage	Gastric Stage	Duodenal Stage	Pronase Stage	Viscozyme L Stage
	*R* ^2^	*R* ^2^	*R* ^2^	*R* ^2^	*R* ^2^
ABTS	0.832	0.818	0.682	0.921	0.907
DPPH	0.860	0.712	0.879	0.731	0.833
FRAP	0.664	0.859	0.887	0.915	0.874

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
