# Peer review of "Colon Bioaccessibility under In Vitro Gastrointestinal Digestion of Different Coffee Brews Chemically Profiled through UHPLC-Q-Orbitrap HRMS"

_foods, 2021, doi:10.3390/foods10010179_

Round 1

Reviewer 1 Report

The authors have studied about bioaccessibility of active compounds from coffee by three brew methods using in vitro digestion model. In addition, CGA and caffeine were measured by ultra-high performance liquid chromatography and ultra-high performance liquid chromatography. This study provided important information about active molecules such as CGA and caffein produced by three different coffee brews and digestion. The goal of study, experimental design and results are clear. The data supports the conclusion of the study. The authors need additional efforts on the manuscript specifically in Discussion section.

Line 22-25: The sentence is too long and containing many different information in one sentence. Split the sentence to clarify.

Line 112-117: Those of three coffee bean samples were same origin (same cultivating location, same production year etc)? If not, the authors need to describe the limitation of this study.

In Discussion section, the authors need to discuss more why simulated gastrointestinal digestion increased bioaccessitility and antioxidant activity of coffee polyphenols compared to non-digested samples. Also, if there is other previously reported information about digestion, and bioaccessitility and antioxidant activity for other type of tea such as green tea or black tea, discuss with the current data.

Author Response

Manuscript ID: foods-1059256

Title: Colon Bioaccessibility under In Vitro Gastrointestinal Digestion of Different Coffee Brews Chemically Profiled through UHPLC-Q-Orbitrap HRMS

Reviewer 1

The authors have studied about bioaccessibility of active compounds from coffee by three brew methods using in vitro digestion model. In addition, CGA and caffeine were measured by ultra-high performance liquid chromatography and ultra-high performance liquid chromatography. This study provided important information about active molecules such as CGA and caffeine produced by three different coffee brews and digestion. The goal of study, experimental design and results are clear. The data supports the conclusion of the study. The authors need additional efforts on the manuscript specifically in Discussion section.

1) Line 22-25: The sentence is too long and containing many different information in one sentence. Split the sentence to clarify.

- As suggested by reviewer 1, the authors split the sentence and reported as: “The main goal of this research was to evaluate the bioaccessibility of phenolic content and variation in antioxidant capacity of three different types of coffee brews after simulated gastrointestinal digestion. This would allow to elucidate how antioxidant compounds present in coffee may exert their effect on the human body, especially in the colonic stage”

2) Line 112-117: Those of three coffee bean samples were same origin (same cultivating location, same production year etc)? If not, the authors need to describe the limitation of this study.

- As suggested by reviewer 1, the authors added the missing information in the manuscript as: As suggested by reviewer 2, the authors added the missing information in the manuscript as: The limitations of our study are mainly related to the diversity of coffee beans used in coffee preparation. Although the espresso and Americano coffee brews were prepared with the same coffee bean samples, the instant coffee samples used in the present study were acquired directly in powder/granules form. This choice was due to the inability to prepare reliable samples from coffee beans that could be comparable to commercially available products in light of the no replicate industrial procedures.

3) In the Discussion section, the authors need to discuss more why simulated gastrointestinal digestion increased bioaccessibility and antioxidant activity of coffee polyphenols compared to non-digested samples. Also, if there is other previously reported information about digestion, bioaccessibility, and antioxidant activity for other types of tea such as green tea or black tea, discuss with the current data.

- As suggested by reviewer 1, the authors added the authors added some discussions in this section: “Similar outcomes were obtained by Annunziata et al., [41] who demonstrated that both colon bioaccessibility and the antioxidant activity of polyphenols from tea were significantly higher compared to the other digestion phases. However, Jilani et al., [42] showed that the black and green tea polyphenols bioaccessibility decreased in the intestinal phase, while the gut microbiota enhanced the total antioxidant activity only in black tea samples. Campos-Vega et al., [43] also demonstrated that colonic fermentation elevates the antioxidant capacity of polyphenols from spent coffee when compared with their respective non-digested sample due to the high release of phenolic compounds. The antioxidant activity was assessed with 2 of 3 assays used in the present work namely the DPPH and ABTS test.”

The authors thank reviewer 1 for evaluating our manuscript.

Reviewer 2 Report

The manuscript entitled ‘Colon Bioaccessibility under In Vitro  Gastrointestinal Digestion of Different Coffee Brews Chemically Profiled through UHPLC-Q-Orbitrap HRMS’ prepared by Castaldo et al. present interesting correlation between type of coffee (espresso, Americano, Instant coffee) and biological (antioxidant) activity as well as colon bioaccessibility.

In my opinion, the Scientists did not altogether think about type of analyzed coffee or the information was not included in the manuscript. In order to obtain reliable results, the same species (i.e. Arabica) of all used coffee should be used . Additionally, the material should be roasted in the same conditions and in ideal studies, the material should purchased form the same provider. In the case of using different types of coffee the obtained results are nonsolid. It is known that secondary plant metabolites can be different  in various species of the same plant what simultaneously  lead to different antioxidant activity.  In my opinion the aforementioned issue should be clarify.  

Additionally, chromatograms obtained for ‘ Identification of Caffeine and CGAs in the Analyzed Coffee Brews Samples Through UHPLC-Q Orbitrap HRMS’ should be provide.

Author Response

Manuscript ID: foods-1059256

Title: Colon Bioaccessibility under In Vitro Gastrointestinal Digestion of Different Coffee Brews Chemically Profiled through UHPLC-Q-Orbitrap HRMS

Reviewer 2

The manuscript entitled ‘Colon Bioaccessibility under In Vitro  Gastrointestinal Digestion of Different Coffee Brews Chemically Profiled through UHPLC-Q-Orbitrap HRMS’ prepared by Castaldo et al. present interesting correlation between type of coffee (espresso, Americano, Instant coffee) and biological (antioxidant) activity as well as colon bioaccessibility.

1) In my opinion, the Scientists did not altogether think about type of analyzed coffee or the information was not included in the manuscript. In order to obtain reliable results, the same species (i.e. Arabica) of all used coffee should be used. Additionally, the material should be roasted in the same conditions and in ideal studies, the material should purchased form the same provider. In the case of using different types of coffee the obtained results are nonsolid. It is known that secondary plant metabolites can be different  in various species of the same plant what simultaneously  lead to different antioxidant activity.  In my opinion the aforementioned issue should be clarify.  

- As suggested by reviewer 2, the authors added the missing information in the manuscript as: As suggested by reviewer 2, the authors added the missing information in the manuscript as: The limitations of our study are mainly related to the diversity of coffee beans used in coffee preparation. Although the espresso and Americano coffee brews were prepared with the same coffee bean samples, the instant coffee samples used in the present study were acquired directly in powder/granules form. This choice was due to the inability to prepare reliable samples from coffee beans that could be comparable to commercially available products in light of the no replicate industrial procedures.

2) Additionally, chromatograms obtained for ‘ Identification of Caffeine and CGAs in the Analyzed Coffee Brews Samples Through UHPLC-Q Orbitrap HRMS’ should be provide.

- As suggested by reviewer 2, total ion chromatogram (TIC) of analyzed samples was added in supplementary materials.

The authors thank reviewer 2 for evaluating our manuscript.

Round 2

Reviewer 1 Report

The authors responded well to the reviewer comments. This version of manuscript is now acceptable.

Reviewer 2 Report

Thank you for improvement the manuscript. Authors improved all required points. In this form the paper is more scientific and readable. In my opinion, it can be  accepted for publication.

This manuscript is a resubmission of an earlier submission. The following is a list of the peer review reports and author responses from that submission.